# Changes in the Volatile Flavor Substances, the Non-Volatile Components, and the Antioxidant Activity of *Poria cocos* during Different Drying Processes

**DOI:** 10.3390/molecules29194777

**Published:** 2024-10-09

**Authors:** Chuqian Gao, Shaodi Sun, Linyu Zhang, Wei Xiang, Miaofen Chen, Jianguo Zeng, Hongqi Xie

**Affiliations:** 1College of Horticulture, Hunan Agricultural University, Changsha 410128, China; outlook_3251813ffff05527@outlook.com (C.G.); weixiang@hunau.edu.cn (W.X.); 2Hunan Key Laboratory of Traditional Chinese Veterinary Medicine, Changsha 410128, China; sunshaodi@stu.hunau.edu.cn (S.S.); zhanglinyu@stu.hunau.edu.cn (L.Z.); miaofen_chen@hunau.edu.cn (M.C.); zengjianguo@hunau.edu.cn (J.Z.); 3College of Veterinary Medicine, Hunan Agricultural University, Changsha 410128, China

**Keywords:** *Poria cocos* (Schw.) wolf, volatile flavor compounds, active ingredients, hot-air drying, shade drying, infrared drying

## Abstract

*Poria cocos* (Schw.) wolf (*P. cocos*) is an important medicinal material with both therapeutic and edible properties. This study investigated volatile constituents, amino acids, proteins, polysaccharides, triterpenoid ingredients, and alcohol-soluble extracts on *P. cocos* during eight drying processes. A total of 47 volatile components were found and identified; the main volatile components of shade drying (SD) and hot-air drying at 50 °C (HD50) were esters and alcohols, while for drying in hot air at 60 °C~100 °C (△ = 10 °C) and infrared drying (ID), the main compounds were aldehydes and hydrocarbons. The amino acids in *P. cocos* remained the same when dried with various methods. Compared with SD samples, with the temperature increase, the content of amino acids showed a trend of first decreasing and then increasing, while the content trend of proteins was the opposite. The HD70 samples had the highest content of polysaccharide, triterpenoid ingredients, alcohol-soluble extracts, and antioxidant activity. Furthermore, volatile compounds showed a correlation between non-volatile constituents. This research provides evidence that the aroma, active components, and activity of *P. cocos* were affected by the drying method.

## 1. Introduction

Shen-nong-ben-cao-jin, the first existing monograph on traditional Chinese medicine in China, is based on the three-category classification and puts forward many pharmaceutical theories [1]. *Poria cocos* (Schw.) wolf (*P. cocos*), initially discovered in the shen-nong-ben-cao-jin, is noted for its anti-inflammatory, antioxidant, anti-tumoral, and hypoglycemic effects, with a clinical compatibility rate of more than 70% [2,3,4]. It is a type of medicinal material with both therapeutic and edible properties, which has important medicinal value as well as a rich nutritional component [5,6]. *P. cocos* is now frequently used in Asian cuisine as a culinary ingredient in meals, teas, porridge, and other nutritious foods.

The drying process is critical as it preserves the quality of the fresh material and prevents bacterial deterioration along the supply chain. At the same time, the Chinese Pharmacopoeia (2020 edition) stipulates that Poria needs to be shade-dried after harvesting, which is a time-consuming and wasteful process. Drying methods have evolved in recent years, including hot-air drying, microwave drying, and others. Each drying method has its own distinct features. In the process of far-infrared drying, the electromagnetic wave energy is directly absorbed by the material, and the heating speed is fast. The principle of hot-air drying is simple and the thermal efficiency is high, which allows it to be used to dry many substances. The drying process of drying in the shade is mild, which can avoid damage from volatile components and pigments in the material caused by high temperature [7]. Zhang discovered that two-stage vacuum and infrared-assisted air impingement drying could improve the quality of *P. cocos* [8]. Zhang revealed that pulse vacuum drying (PVD) technology improves the drying efficiency and quality of *P. cocos* [9].

The chemical composition of *P. cocos* includes polysaccharides, triterpenoids, proteins, amino acids, and other minor families of molecules [10,11,12]. Polysaccharides, triterpenoids, and amino acids were identified as the most important group of bioactive components [13]. Studies have shown that *P. cocos* alkali-soluble polysaccharide has anti-inflammatory and immunomodulatory effects [14]. The triterpene compound pachymic acid (PA) in *P. cocos* can regulate lipid metabolism and be anti-tumor [15,16]. In addition, amino acids are very essential for the human body. At the same time, several recent studies have shown that aroma is also one of the food quality determinants [17,18]. Surprisingly, no research has been conducted to evaluate the changes in proteins, amino acids, active ingredients, and smell during the *P. cocos* drying process. This study examined the dynamic changes in volatile compounds and non-volatile components of *P. cocos* during eight drying processes, analyzed the interrelationships between these compounds, investigated the mechanisms of action of the relevant compounds in *P. cocos*, synthesized the antioxidant activity effects, and then identified the best method for drying *P. cocos*. The main objective of this study was to provide data support and practical guidance to help with the selection of optimum drying techniques for *P. cocos* and improve the general quality of the product.

## 2. Results and Discussion

### 2.1. Volatile Flavor Compound (VFC) Analysis of P. cocos

#### 2.1.1. Analysis of Volatile Compounds Identified by GCMS

Volatile flavoring components influence appetite, which in turn influences intake and digestion, and there exists a study that has found that the changes in the volatile fraction also affect the efficacy of herbs [19]. Therefore, to understand the volatile compound composition of *P. cocos* during different drying processes, the types and contents of volatile compounds in *P. cocos* were detected using HS-SPME-GCMS under eight drying procedures, as shown in Appendix A (Appendix A). A total of 47 compounds were detected in the *P. cocos*, comprising 9 aldehydes, 16 hydrocarbons, 6 alcohols, 12 esters and ketones, and 4 others. The identification of volatile chemicals was generally consistent with the findings of the following research [20].

Specifically, it was discovered that when the drying temperature was gradually increased, the relative alcohol and ester contents of *P. cocos* dried at HD 100 was reduced by 70% and 75% dividedly compared to SD, which may be because alcohols and esters were abundant in fresh *P. cocos* but were highly volatile and unstable during heating [21].The relative concentration of aldehydes and hydrocarbons increased with increasing drying temperature, by 53% and 208%, respectively. They are the primary results of the Maillard process, with green, nutty, sweet, flowery, and woody flavors [22]. Among them, D-limonene is a substance with antioxidant and anti-inflammatory properties. The significant increase in its content is likely to have a positive impact on the antioxidant activity of *P. cocos* [23,24]. The increase in the content of 2-undecanone, a volatile component of ketones, and its role in reducing myocardial inflammatory injury indicate that *P. cocos* has great potential for cardiac protection [25]. The relative concentrations of hexanal and tetradecane increased by 342% and 184%, respectively. These chemicals could have been generated via a martensitic reaction [26]. As a result, a martensitic reaction was clearly observed. Furthermore, Table 1 examines the odor of 47 VFC components, demonstrating that aldehydes, hydrocarbons, and alcohols have an important influence on the flavor quality of *P. cocos*.

Figure 1 shows the radar chart of VFCs at different drying processes. The main components of SD and HD50 were esters and alcohols, while the other six drying methods’ main compounds were aldehydes and hydrocarbons. Given that, an increase in the content of α-terpineol, a volatile component of honeyed ephedra, may be associated with enhanced asthma-calming effects. Therefore, different drying methods may also affect the efficacy of the herb.

#### 2.1.2. PCA, HCA, and Pearson’s Correlation Analyses of Volatile Flavor Compounds

The impact of the drying process on the distribution of volatile chemicals in *P. cocos* was investigated using PCA. Figure 2A illustrates that the first and second significant components account for 76.2% of the volatile compounds (PC1: 47.7%; PC2: 19.5%). The results of the PCA in Figure 2A demonstrate that the PC1 and PC2 scores can be used to discriminate between samples at different stages of drying. The shade-dried samples were identified using the PC2 score. The remaining seven drying stages were organized in clusters. Based on the PCA results in Figure 2B, the 47 volatile compounds can be divided into numerous clusters. The compounds were not spread uniformly. The samples were identified based on their similarity in volatile chemicals (Figure 2C). The samples were separated into three groups based on their horizontal drying stages: shade-dried, hot air-dried at 90–100 °C (Δ = 10 °C), hot air-dried at 50–80 °C (Δ = 10 °C), and infrared-dried. This indicates that drying temperature can obviously affect volatile compounds. A total of 47 compounds were vertically classified into two clusters, the first of which contained the compounds Nonanal/(E)-, 2-Nonenal/Decanal/Undecanal/Dodecanal/Cyclohexane, 1-ethenyl-1-methyl-2-(1-methylethenyl)-4-(1-methylethylidene)-/1-Nonanol/(E)-, 1,6,10-Dodecatrien-3-ol, 3,7,11-trimethyl-/Cyclopropane methanol, α.,2-dimethyl-2-(4-methyl-3-pentenyl)-, [1.α.(R*),2. α.]-/Nonanoic acid/Butanoic acid, butyl ester/Nonanoic acid, 9-oxo-, methyl ester/Pentadecanoic acid, methyl ester/1,2-Benzenedicarboxylic acid, bis(2-methyl propyl) ester/Hexadecanoic acid, methyl ester/Methyl linoleate/Isopentyl 3,5,5-trimethyl hexanoate/5,9-Undecadien-2-one, 6,10-dimethyl-/2-Undecanone/γ-Muurolene; the other 27 volatile compounds were divided into the second cluster.

A Pearson correlation analysis was used to find the correlation coefficients of the volatile chemicals in the drying process. Figure 2D displays the correlations between the 47 volatile components. Positive and negative correlations are represented by the colors red and blue, respectively. Except for 3,5-Octadien-2-ol/Cyclohexane, 1-ethenyl-1-methyl-2,4-bis(1-methyl ethenyl)-, [1S-(1α,2*β*,4*β*)]-/Cyclohexane, 1-ethenyl-1-methyl-2-(1-methyl ethenyl)-4-(1-methyl ethylidene)-/Linalool/2,2,4-Trimethyl-1,3-pentanediol diisobutyrate/2-Undecanone/5,9-Undecadien-2-one, 6,10-dimethyl-/Benzene, [[(1-ethenyl-1,5-dimethyl-4-hexenyl) oxy] methyl]-, which show no significant correlation with other volatile compounds, compounds **28**–**41** and **44**–**46** showed significant negative correlation with volatile compounds **1**–**26** and **28**–**40**, respectively.

In contrast, the compounds Hexanal/Decanal/*β*-Eudesmene/1,6,10-Dodecatrien-3-ol, 3,7,11-trimethyl-, (E)-/Cyclopropane methanol, α,2-dimethyl-2-(4-methyl-3-pentenyl)-, [1. α.(R*),2. α.]-/Nonanoic acid/Butanoic acid, butyl ester/Pentadecanoic acid, and methyl ester show a strong positive correlation with many other compounds. The aforementioned statistical studies’ findings give essential information for analyzing volatile chemicals in *P. cocos* during the drying process and partially reflect the variations in volatile compounds in the material at different stages of drying.

### 2.2. Changes in Amino Acids (AA), Crude Proteins (CP), and Water-Soluble Proteins (WSP) during Different Drying Processes

Table 1 shows that the eight drying methods had no impact on amino acid classification. All types of amino acids showed the same pattern of change: increasing temperature led to a decrease in amino acid content. The total amino acid content decreased in the following order across all drying techniques: HD50, HD90, HD100, HD80, HD60, ID, SD, and HD70, with a significant difference (*p* < 0.05). The total amino acids content varied from 1.16 ± 0.058 to 3.52 ± 0.804 mg/g dw.

From high to low, the major three amino acids in the SD were Asp, Glu, and Leu, and the first three amino acids in the HD50 were the same as those in the SD. However, throughout the entire drying process of the HD (60~100 °C, Δ = 10 °C) and ID samples, the first three amino acids were Glu, Val, and Leu. The total amino acids content of the six hot-air drying procedures dropped before increasing (Figure 3A), which was consistent with earlier findings [27]. Compared with the SD samples, except the HD70, the total amino acid content of the other six drying methods was increased significantly (*p* < 0.05). The trend for *CP* and *WSP* content was opposite to that of amino acids, showing an increasing and then decreasing trend. These changes can be explained by the fact that during the drying process, as the temperature increases, enzyme activity slows down and protein hydrolysis decreases, resulting in a downward trend in total amino acid content. After 70 °C, the total amino acid content increased and the protein content decreased with increasing temperature, which may be related to the degradation and lower solubility of proteins at high temperatures.

### 2.3. Changes in ASP, TTD, ASE, and Other Bioactive Components during Different Drying Processes

As shown in Table 2, the *ASP* contents of HD (50~100 °C, △ = 10 °C), SD, and ID samples were 234.08 ± 14.78, 327.94 ± 3.23, 400.68 ± 3.05, 375.35 ± 2.76, 366.64 ± 8.55, 354.89 ± 1.10, 213.32 ± 25.61, and 238.15 ± 4.83, respectively, with significant differences (*p* < 0.05). The *ASP* content of the HD50 and ID samples is not significantly different from that of the SD samples. However, the *ASP* content of the HD (60~100 °C, △ = 10 °C) samples differs significantly (*p* < 0.05). As the temperature rises, the content of *ASP* displays an increasing and subsequently decreasing tendency, which is consistent with the results of earlier investigations [28]. The significant decrease (*p* < 0.05) in the *ASP* content of *P. cocos* was related to its high-temperature degradation.

Triterpenoids are the most common terpenoids in *P. cocos*, and they are one of its primary active constituents. In Table 2, the *TTD* content of the HD (50~100 °C, △ = 10 °C), SD, and ID samples was 5.43 ± 0.03, 5.95 ± 0.03, 6.84 ± 0.03, 5.15 ± 0.07, 4.85 ± 0.05, 4.90 ± 0.04, 5.95 ± 0.46, and 4.94 ± 0.23. Aside from HD70, there were no significant differences between the other six drying techniques and SD samples. The content of *TTD* shows an increasing and then declining pattern with temperature, which may be because an optimum drying temperature can offer a good environment for enzyme activities of cells at a stage of high moisture content, thereby elevating the triterpenoid content [29].

The *ASE* content is a vital index in the Chinese Pharmacopoeia. There was no significant difference in *ASE* content among the eight drying processes, but all fulfilled the pharmacopeia standard of more than 2.5% and were of good quality. It is worth noting that the observed changes in the *ASE* during the drying process were identical to those in the *ASP* and *TTD*. In contrast, variations in the content of the six triterpenoids followed the reverse pattern, but with no significant differences.

### 2.4. PCA, HCA, and Pearson’s Correlation Analyses of AA, CP, WSP, ASP, TTD, Other Bioactive Components, and ASE

As shown in Figure 4A, the results can be divided into two quadrants as follows: the shade drying, HD50, and ID samples were grouped together, while the other five drying samples were clustered together. Figure 4B illustrates the PCA results of the amino acids, two proteins, alkali-soluble polysaccharide, total triterpenoid, triterpenoids, and alcohol soluble extracts, which were roughly divided into two clusters.

Figure 3C shows two horizontal clusters: the first cluster is HD70, while the second cluster includes HD50, HD60, HD (80~100 °C, ∆ = 10 °C), SD, and ID. The vertical direction is separated into four clusters: the first cluster comprises PA, DTE-1, PAC, and DTA-2; the second cluster includes all amino acids except Cys; the third cluster includes *ASP* and *ASE*; and the fourth cluster contains *TTD*, EA-1, Cys, *CP*, and *WSP*.

Figure 4D depicts the correlation analysis results, which demonstrate the positive and negative connections between amino acids, proteins, ASP, TTD, and ASE. Figure 4D shows that PAC had a substantial positive connection with His, while DTA-2 had a positive association with His, Pro, Ala, Arg, Glu, and Asp. The remaining higher content components had a significant negative correlation with amino acids. The above results indicated that the amino acids and some chemical components in *P. cocos* would change during the drying process, and the correlation between the amino acids and other components of *P. cocos* could provide a theoretical foundation for future research into the changes in *P. cocos* internal chemical components.

**Figure 4 molecules-29-04777-f004:**
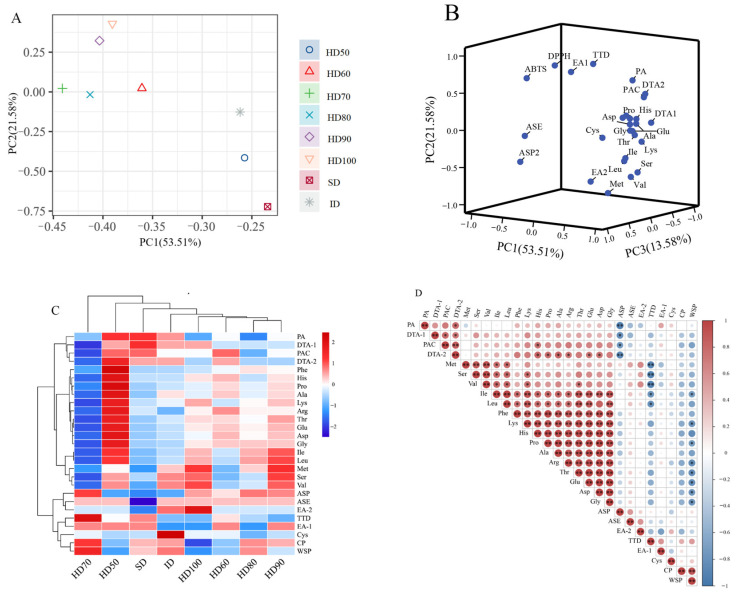
Principal component analysis, hierarchical cluster analysis, and Pearson’s correlation analysis based on the contents of the *AA*, *CP*, *WSP*, *ASP*, *TTD*, other bioactive components, and ASE in the samples of the eight drying processes. The abbreviated words represent the non-volatile flavor compounds in Table 2. (**A**) Score plot of the principal component analysis of the different drying methods; (**B**) score plot of the principal component analysis; (**C**) heatmap of the hierarchical cluster analysis; (**D**) heatmap of Pearson’s correlation analysis. * *p* < 0.05, ** *p* < 0.01.

### 2.5. Antioxidant Activities

Since many human diseases are related to oxidation, the interest in research on antioxidant properties is increasing [30]. This study assessed the antioxidant activity of *Poria cocos* powder extracts using two popular techniques (DPPH and ABTS^+^) (Table 2). The HD70 samples had the highest antioxidant activity with both assays, followed by HD60. The ID-treated *P. cocos* had the lowest antioxidant activity with DPPH and ABTS^+^. The HD70 samples had considerably increased antioxidant activity compared to the other drying processes (*p* < 0.05) in DPPH. According to Figure 3B,C and Table 2, this phenomenon may be closely related to the content of *ASP*, *TTD*, and protein [31,32]. The low temperature of HD can efficiently inhibit the oxidation and breakdown of *ASP* and *TTD*, while also inactivating the degradative enzymes more quickly. However, the ABTS^+^ antioxidant activity did not differ between the HD50, HD60, HD80, and SD samples, possibly due to differing detection principles [33,34].

### 2.6. Pearson’s Correlation Analyses of Volatile Components, AA, CP, WSP, ASP, TTD, Other Bioactive Components, ASE, and Antioxidant Activity

In order to know the relationship between the volatile components and the non-volatile components, the correlation was further analyzed, as shown in Figure 5. Met, Ile, Leu, Pro, Phe, Dta-1, PA, ASE, DTA-2, TTD, Asp, Glu, Gly, His, Arg, and ASP showed strong negative correlations with V1, V2, V4, V6, V7, V10, V11, V14, V15, V28, V30, V31, V32, V33, V35, V36, V38, V39, V40, V41, and V42, respectively. Among them, V42 had a fruity flavor but was inversely associated with many amino acids, implying that the more amino acids generated, the lower the V42 concentration. The volatile components of V37 were strongly and positively connected with Pro, Ala, Phe, and DTA-2, while the other volatile components had no significant link with the non-volatile components. The interrelationship between the volatile and non-volatile compounds suggests that some flavor compounds are affected by other components during drying, which could provide theoretical support for processing *P. cocos* with good flavor.

## 3. Materials and Methods

### 3.1. Materials

Fungal materials: the fresh *P. cocos* was purchased from Jingzhou Zhimei Health Technology Co., Ltd. (Jingzhou, China) and authorized by Prof. Zeng Jianguo, Post Scientist of Harvesting and Preliminary Processing of National Chinese Materia Medica Industry Technology System, Hunan Agricultural University. Fresh *P. cocos* were steamed “sweated” for 3 h; then, cut into *P. cocos* blocks; and dried in hot air (50 °C~100 °C, △ = 10 °C), the shade, and far infrared, respectively. All processed samples were pulverized, passed through a 60-mesh sieve, and stored in sealed bags at constant temperature in a Chinese herbal medicine storage cabinet.

### 3.2. Determination of Volatile Compounds by GC-MS

To determine the volatile chemicals, 1.000 ± 0.005 g of the test sample was placed in a 5 mL volumetric flask and immersed in a water bath at 70 °C for 40 min. Then, the SPEM fibers were exposed to the upper space of the flask and immersed for 40 min at 70 °C. The SPEM fibers were then removed and placed in the GC-MS injector at 250 °C for 5 min of desorption.

The volatile chemicals were detected using a GC-MS coupled with a DB-5MS capillary column (30 m × 250 μm, 0.25 μm). The heating procedure began with a temperature of 50 °C for 5 min and then increased to 130 °C at a rate of 4 °C/min for 2 min, and finally to 230 °C at a rate of 4 °C/min for 3 min. The total analysis duration was 54 min. The inflow temperature remained at 250 °C. The spitless injection method was used. The EI ion source was employed at 230 °C. The scanning mode was full scan, and the scanning period was 5–54 min. The mass-to-charge ratio scanning range was 45–500 *m*/*z*.

The volatile components detected by GC-MS were quantitatively compared using a computer and the NIST14 and NIST17 standard spectral libraries, and the results with a similarity above 80 were chosen as the identification result. The controls were *P. cocos* samples that had been dried in the shade. The peak area normalization approach was utilized in a quantitative analysis.

### 3.3. Determination of Amino Acids (AAs)

The amino acid concentration of the material after processing was evaluated with an automatic amino acid analyzer (L-8900, Hitachi, Japan). Firstly, the dried samples were weighed precisely and placed in glass tubes. Then, 10 mL of dilute hydrochloric acid (1:1) was poured into a sealed glass tube and heated at 110 °C for 24 h for hydrolysis. Following hydrolysis, the solution was carefully transferred to a 25 mL volumetric flask and diluted to the mark with deionized water. The filtrate was then filtered and evaporated in a water bath at 95 °C. Then, 1 mL of deionized water was added and evaporated. The solution was then accurately transferred into a 10 mL volumetric vial and dissolved with dilute hydrochloric acid (1:600 *v*/*v*), which was diluted to the scale line. Lastly, the solution was filtered through a 0.45 µm membrane to ascertain the component content. The automatic amino acid analyzer uses ion exchange resins and ninhydrin colorimetry to measure the amount of each chemical by measuring the absorbance. The data were then produced using a photometer [35].

### 3.4. Determination of Crude Protein (CP)

The crude protein composition of raw material samples was determined using an automatic Kjeldahl tester (VAP450 + KT20s, Bonn, Germany). Firstly, 2.5 g of sample powder was weighed precisely and placed in a digestion tube, and then, the catalyst CuSO_4_ 0.4 g, K_2_SO_4_ 6.0 g, and concentrated H_2_SO_4_ 12 mL were added; put into the digestion box; and then programmed to be heated up at 410 °C for 180 min. Finally, the amount of *CP* was calculated using the amount of hydrochloric acid absorbed by the Kjeldahl automatic nitrogen detector.

### 3.5. Determination of Water-Soluble Protein (WSP)

An amount of 1.0 g of sample powder was precisely weighed and placed in a conical flask. Then, 5 mL of deionized water was added, followed by 30 min sonication (200 W, 40 kHz). The sample solution was thoroughly filtered with a 0.45 µm filter. The experiment was then conducted per the TP test kit instructions to determine the water-soluble protein concentration. The absorbance of the post-reaction solution was measured at 595 nm.

### 3.6. Determination of Alkali-Soluble Polysaccharide (ASP)

An amount of 0.5 g of dry material was weighed carefully, and 5.0 mL of 1.0 M NaOH solution was added and then sonicated (200 W, 40 kHz) at room temperature for 30 min. The filtrate was then passed through a 0.45 µm filter. For colorimetric analysis, the sample solution was 100-fold diluted with NaOH folding, followed by the addition of 2.0 mL of 5% (*w*/*v*) phenol and 5.0 mL of H_2_SO_4_ to 1 mL of the extract. The mixture was properly combined, heated in a boiling water bath for 15 min, and then quickly cooled to room temperature using ice. The absorbance of the solution following the reaction was measured at 490 nm.

### 3.7. Determination of Total Triterpenoid (TTD)

Then, 1.0 g of sample powder was weighed precisely and placed in a conical flask; 10 mL of 100% methanol was added and then sonicated for 30 min at room temperature (200 W, 40 kHz). To perform a colorimetric analysis, 1 mL of the solution was carefully put into a centrifuge tube, and the solvent was evaporated in a water bath and chilled to room temperature, followed by 0.2 mL of 5% (*w*/*v*) vanillin-glacial acetic acid solution and 0.8 mL of perchloric acid being added sequentially. After properly mixing the mixture, it was treated in a constant temperature water bath at 70 °C for 20 min and promptly cooled to room temperature with ice. Then, 5 mL of glacial acetic acid was added and shaken well. The absorbance of the post-reaction solution was measured at 560 nm.

### 3.8. Determination of Six Triterpenoid Ingredients

#### 3.8.1. Extraction

Then, 1.0 g of dried samples was sonicated (200 W, 40 kHz) at ambient temperature for 30 min in 10 mL of 100% methanol and then filtered, and the filtrate was collected. Next, the filtrate was passed through a 0.45 µm filter after thoroughly mixing to obtain the sample solution.

#### 3.8.2. HPLC Condition

High-Pressure Liquid Chromatography (HPLC) was performed by following the method published in our previous work [36]. Chromatography separation was achieved utilizing the 51 min gradient delivery of a mixture of ultrapure water (A) and HPLC grade acetonitrile supplied by Sinopharm Chemical Reagent Co., Ltd. (Shanghai, China), each containing 0.2% of HPLC grade acetic acid (B), flow rate: 1.0 mL/min. The mobile phase was composed of water/acetonitrile (*v*/*v*) with a gradient elution: 0~5 min, 45~50% (B); 5~45 min, 50~90% (B); 45~46 min, 90~45% (B); and 46~51 min, 45~45% (B). An Agilent-ZORBAX SB-C18 (250 mm × 4.6 mm, 5 μm, Beijing, China) column was applied, and the column temperature was 30 °C.

### 3.9. Alcohol Soluble Extract (ASE)

The *ASE* content of herbal medications, which includes components such as triterpenoids, polysaccharides, steroids, and phenols, is a crucial predictor of their overall quality [37]. The *ASE* content was determined using the hot socking method, as stated in the Chinese Pharmacopoeia (2020).

### 3.10. Antioxidant Capacity

#### 3.10.1. Preparation of Extracts

The extraction procedure was similar to the prior method, with minor changes [38]. The dry powdered sample (60 g) was sonicated (200 W, 40 kHz) with 80% ethanol (1:10 *w*/*v*) for 1 h at ambient temperature before being filtered. The extraction was repeated with an additional volume of 80% ethanol for 1 h, and the combined filtrates were reduced to one-third of the original volume by rotary evaporation. After drying in hot air at 60 °C, the crude extract powder was kept in a glass jar at 20 °C.

#### 3.10.2. Determination of Antioxidant Activity

The antioxidant capacity was measured by DPPH and ABTS. The antioxidant capacities of the eight drying samples were measured by using the ABTS and DPPH kits (20230706(A015-2-1), 20240301(A153-1-1), Nanjing Jiancheng Bioengineering Institute, Nanjing, China).

### 3.11. Data Analysis

The experimental results were analyzed using ANOVA and Tukey’s test in SPSS 23.0, with *p* < 0.05 indicating statistical significance. A principal component analysis (PCA), a hierarchical cluster analysis (HCA), and Pearson’s correlation analysis were all carried out with Origin 2022 and SPSS 23.0.

## 4. Conclusions

In this study, we investigated the dynamics and correlations of volatile compounds with non-volatile compounds in *P. cocos* during drying by hot air (50~100 °C, △ = 10 °C), shade drying and infrared drying. A total of 47 compounds were detected in the *P. cocos* using GC-MS. Compared with SD, the subtotal content of aldehydes, hydrocarbons and other volatile compounds shows a significant (*p* < 0.05) increase, and the subtotal content of alcohols, esters, and ketones shows a significant (*p* < 0.05) decrease. The highest total amino acid content was found in the HD50 sample, and the lowest total amino acids content was found in the HD70 sample; however, the highest content of protein was found in HD70 and the lowest total protein content was found in HD50, with a significant difference. HD70 samples have the highest content of *ASP*, *TTD*, *ASE*, and antioxidant activity (DPPH and ABTS^+^), 400.68 ± 3.05, 6.84 ± 0.03, 3.07 ± 0.003 mg/g, and 33.06 ± 1.52, 32.72 ± 1.01 (mg trolox/g extract), respectively, and the content of the six triterpenoids have no significant difference during different drying processes. The correlation analysis showed that there was an association between volatile flavor substances and non-volatile components. Based on these findings, it can be stated that the HD70 method is suitable for drying *P. cocos*.

## Figures and Tables

**Figure 1 molecules-29-04777-f001:**
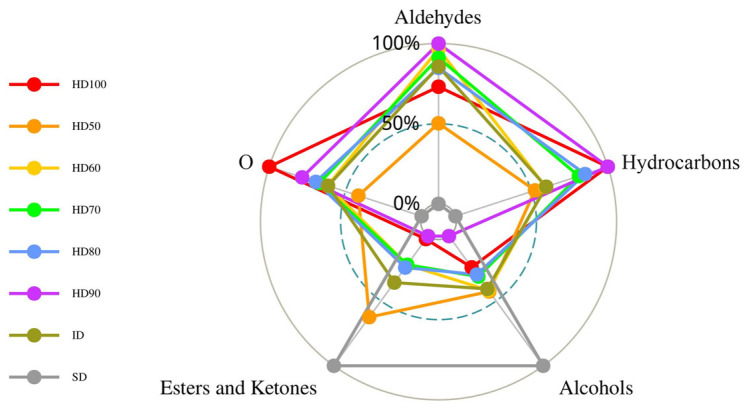
Radar chart of VFCs at different drying processes.

**Figure 2 molecules-29-04777-f002:**
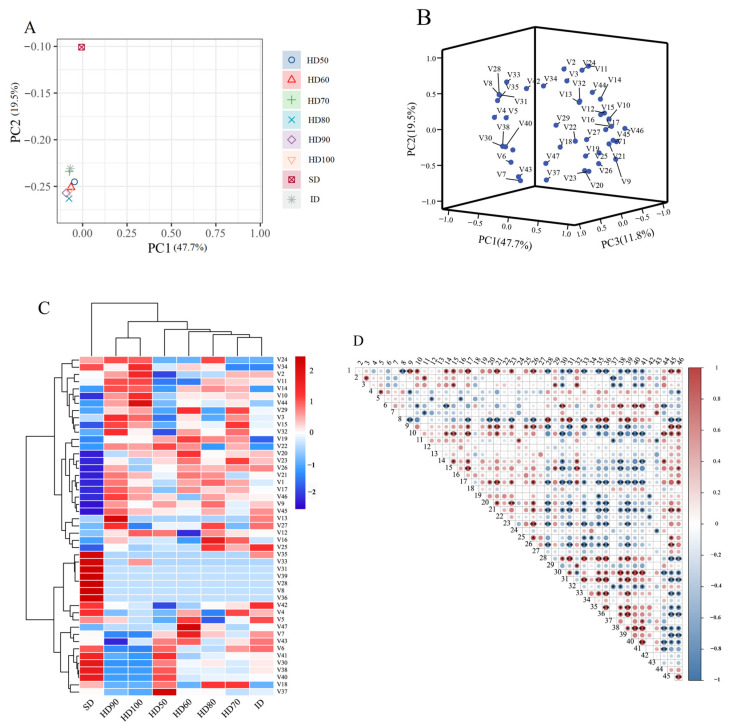
Principal component analysis, hierarchical cluster analysis, and Pearson’s correlation analysis based on the contents of the volatile flavor compound in the samples of the eight drying processes. The abbreviated words represent the non-volatile flavor compounds in Table 1. (**A**) Score plot of the principal component analysis of the different drying methods; (**B**) score plot of the principal component analysis; (**C**) heatmap of the hierarchical cluster analysis; (**D**) heatmap of Pearson’s correlation analysis. * *p* < 0.05, ** *p* < 0.01.

**Figure 3 molecules-29-04777-f003:**
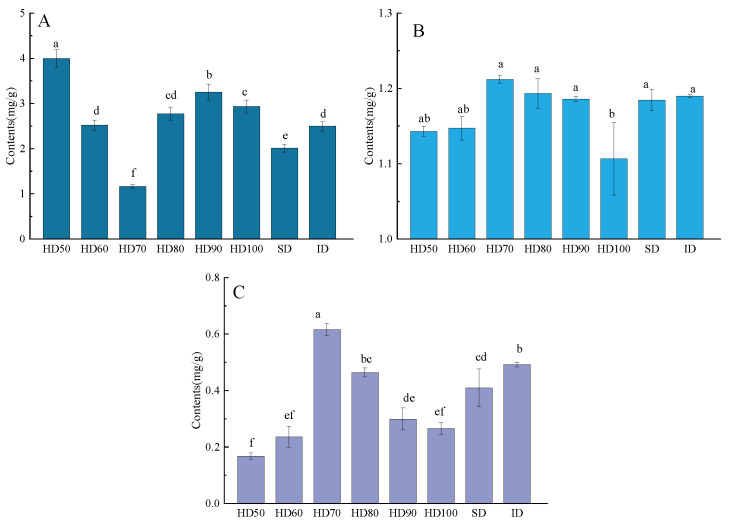
The content of total amino acids (**A**), the content of crude proteins (**B**), and the content of water-soluble protein (**C**) in different drying methods. Different letters (a, b, c, d…) mean significantly different (*p* < 0.05) as determined by one-way analysis of variance (ANOVA), followed by Tukey’s test.

**Figure 5 molecules-29-04777-f005:**
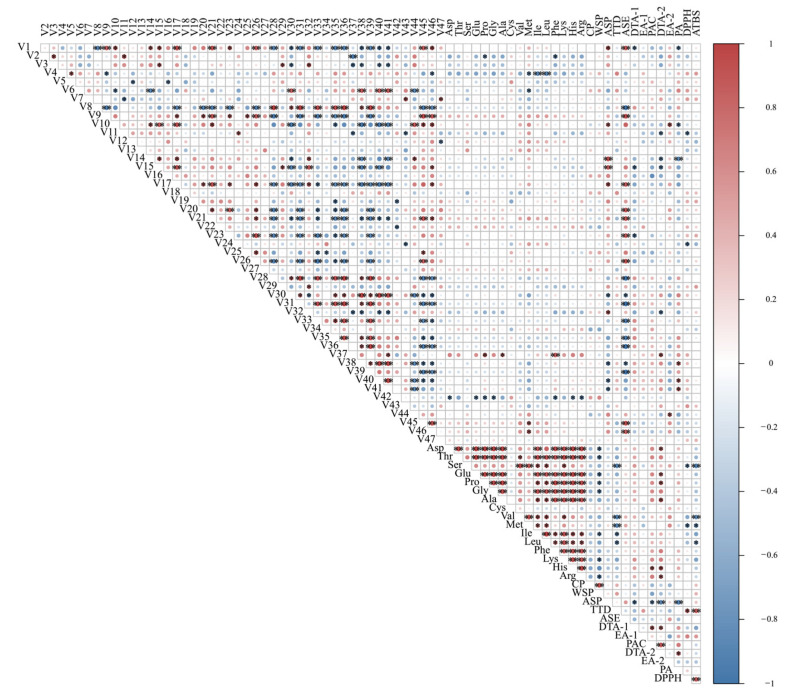
Heatmap of Pearson’s correlation analysis. * *p* < 0.05, ** *p* < 0.01.

**Table 1 molecules-29-04777-t001:** Contents of amino acids and proteins in the *Poria cocos* samples during drying.

Compound Category	Compound Name	Content/(mg/g)							
		HD50	HD60	HD70	HD80	HD90	HD100	SD	ID
Amino Acids	Asp	0.41 ± 0.007 ^a^	0.25 ± 0.000 ^b^	0.13 ± 0.000 ^e^	0.26 ± 0.006 ^bc^	0.29 ± 0.015 ^b^	0.25 ± 0.000 ^c^	0.21 ± 0.000 ^d^	0.23 ± 0.021 ^cd^
	Ser	0.15 ± 0.007 ^a^	0.11 ± 0.001 ^b^	0.08 ± 0.006 ^c^	0.12 ± 0.006 ^b^	0.17 ± 0.012 ^a^	0.16 ± 0.005 ^a^	0.12 ± 0.001 ^b^	0.15 ± 0.006 ^a^
	Glu	0.43 ± 0.007 ^a^	0.30 ± 0.002 ^b^	0.15 ± 0.005 ^d^	0.28 ± 0.006 ^b^	0.31 ± 0.023 ^b^	0.28 ± 0.006 ^b^	0.22 ± 0.006 ^c^	0.24 ± 0.020 ^c^
	Pro	0.15 ± 0.035 ^a^	0.10 ± 0.012 ^b^	0.06 ± 0.006 ^d^	0.09 ± 0.002 ^bc^	0.10 ± 0.010 ^b^	0.09 ± 0.010 ^bc^	0.08 ± 0.008 ^c^	0.09 ± 0.012 ^bc^
	Gly	0.15 ± 0.040 ^a^	0.10 ± 0.040 ^b^	0.03 ± 0.000 ^d^	0.10 ± 0.001 ^b^	0.10 ± 0.010 ^b^	0.09 ± 0.006 ^bc^	0.07 ± 0.000 ^c^	0.07 ± 0.006 ^c^
	Ala	0.28 ± 0.067 ^a^	0.20 ± 0.020 ^b^	0.10 ± 0.001 ^d^	0.17 ± 0.002 ^bc^	0.19 ± 0.017 ^b^	0.17 ± 0.001 ^bc^	0.15 ± 0.006 ^c^	0.19 ± 0.025 ^b^
	Cys	0.03 ± 0.010 ^a^	0.03 ± 0.001 ^a^	0.03 ± 0.001 ^a^	0.02 ± 0.010 ^a^	0.03 ± 0.006 ^a^	0.03 ± 0.010 ^a^	0.03 ± 0.006 ^a^	0.06 ± 0.049 ^a^
	His	0.06 ± 0.030 ^a^	0.04 ± 0.012 ^b^	0.01 ± 0.006 ^c^	0.03 ± 0.001 ^b^	0.04 ± 0.006 ^b^	0.04 ± 0.006 ^b^	0.03 ± 0.001 ^b^	0.03 ± 0.002 ^b^
	Arg	0.08 ± 0.023 ^a^	0.06 ± 0.017 ^b^	0.02 ± 0.001 ^f^	0.05 ± 0.005 ^c^	0.05 ± 0.004 ^c^	0.05 ± 0.002 ^c^	0.04 ± 0.006 ^d^	0.03 ± 0.006 ^e^
	Thr	0.21 ± 0.046 ^a^	0.15 ± 0.035 ^b^	0.07 ± 0.001 ^e^	0.14 ± 0.011 ^bc^	0.16 ± 0.012 ^b^	0.15 ± 0.010 ^b^	0.12 ± 0.006 ^d^	0.13 ± 0.010 ^cd^
	Val	0.34 ± 0.010 ^a^	0.25 ± 0.032 ^bc^	0.15 ± 0.001 ^d^	0.26 ± 0.026 ^b^	0.37 ± 0.015 ^a^	0.38 ± 0.015 ^a^	0.20 ± 0.030 ^cd^	0.34 ± 0.012 ^a^
	Met	0.13 ± 0.029 ^ab^	0.09 ± 0.012 ^b^	0.07 ± 0.006 ^b^	0.14 ± 0.044 ^ab^	0.19 ± 0.035 ^a^	0.18 ± 0.023 ^a^	0.07 ± 0.031 ^b^	0.14 ± 0.025 ^ab^
	Ile	0.27 ± 0.075 ^a^	0.16 ± 0.038 ^a^	0.06 ± 0.006 ^c^	0.21 ± 0.059 ^a^	0.25 ± 0.044 ^a^	0.21 ± 0.044 ^ab^	0.12 ± 0.006 ^b^	0.15 ± 0.017 ^b^
	Leu	0.77 ± 0.017 ^a^	0.43 ± 0.107 ^c^	0.14 ± 0.023 ^d^	0.63 ± 0.044 ^ab^	0.75 ± 0.078 ^a^	0.61 ± 0.010 ^b^	0.34 ± 0.038 ^c^	0.43 ± 0.050 ^c^
	Phe	0.40 ± 0.188 ^a^	0.17 ± 0.071 ^bcd^	0.05 ± 0.006 ^e^	0.19 ± 0.017 ^b^	0.18 ± 0.015 ^bc^	0.14 ± 0.010 ^bcd^	0.11 ± 0.00 ^de^	0.12 ± 0.017 ^cd^
	Lys	0.13 ± 0.029 ^a^	0.08 ± 0.021 ^b^	0.03 ± 0.006 ^d^	0.08 ± 0.006 ^bc^	0.09 ± 0.006 ^b^	0.10 ± 0.006 ^b^	0.07 ± 0.006 ^c^	0.08 ± 0.010 ^bc^
	Subtotal	4.00 ± 0.804 ^a^	2.52 ± 0.5782 ^d^	1.16 ± 0.058 ^f^	2.77 ± 0.220 ^cd^	3.25 ± 0.310 ^b^	2.93 ± 0.135 ^c^	1.98 ± 0.150 ^e^	2.50 ± 0.292 ^d^
Protein	CP	1.14 ± 0.069 ^ab^	1.15 ± 0.016 ^ab^	1.21 ± 0.005 ^a^	1.19 ± 0.020 ^a^	1.19 ± 0.0035 ^a^	1.11 ± 0.048 ^b^	1.18 ± 0.015 ^a^	1.19 ± 0.002 ^a^
	WSP	0.17 ± 0.115 ^f^	0.24 ± 0.038 ^ef^	0.62 ± 0.356 ^a^	0.46 ± 0.268 ^bc^	0.30 ± 0.175 ^de^	0.26 ± 0.154 ^ef^	0.41 ± 0.066 ^cd^	0.49 ± 0.008 ^b^

Note: Duncan’s labeling method was used between the treatment groups. Peer data with the same letter (a, b, c, d…) show no significant difference or negligible difference across treatment groups (*p* > 0.05), while excluding the same letter indicates a significant difference (*p* < 0.05). HD means hot-air drying, SD means integrated shade drying, and ID means conventional infrared drying.

**Table 2 molecules-29-04777-t002:** Contents of *ASP*, *TTD*, *ASE*, other bioactive components, and antioxidant activity in the *Poria cocos* samples during drying.

Compound Name	Content							
	HD50	HD60	HD70	HD80	HD90	HD100	SD	ID
ASP/(mg/g)	234.08 ± 14.78 ^d^	327.94 ± 3.23 ^c^	400.68 ± 3.05 ^a^	375.35 ± 2.76 ^ab^	366.64 ± 8.55 ^b^	354.89 ± 1.10 ^bc^	213.32 ± 25.61 ^d^	238.15 ± 4.83 ^d^
TTD/(mg/g)	5.43 ± 0.03 ^b^	5.95 ± 0.03 ^b^	6.84 ± 0.03 ^a^	5.15 ± 0.07 ^b^	4.85 ± 0.05 ^b^	4.90 ± 0.04 ^b^	5.95 ± 0.46 ^b^	4.94 ± 0.23 ^b^
ASE/%	2.82 ± 0.003 ^a^	2.94 ± 0.003 ^a^	3.07 ± 0.003 ^a^	2.96 ± 0.007 ^a^	2.89 ± 0.001 ^a^	2.72 ± 0.005 ^a^	2.50 ± 0.003 ^a^	2.74 ± 0.001 ^a^
DTA-1/(mg/g)	0.30 ± 0.02 ^a^	0.29 ± 0.08 ^a^	0.27 ± 0.03 ^a^	0.29 ± 0.01 ^a^	0.29 ± 0.01 ^a^	0.30 ± 0.01 ^a^	0.31 ± 0.01 ^a^	0.30 ± 0.03 ^a^
EA-1/(mg/g)	0.09 ± 0.01 ^a^	0.09 ± 0.02 ^a^	0.09 ± 0.01 ^a^	0.08 ± 0.00 ^a^	0.09 ± 0.03 ^a^	0.08 ± 0.00 ^a^	0.09 ± 0.00 ^a^	0.08 ± 0.00 ^a^
PAC/(mg/g)	0.22 ± 0.01 ^a^	0.22 ± 0.06 ^a^	0.19 ± 0.02 ^a^	0.20 ± 0.01 ^a^	0.21 ± 0.06 ^a^	0.21 ± 0.01 ^a^	0.22 ± 0.01 ^a^	0.21 ± 0.02 ^a^
DTA-2/(mg/g)	0.22 ± 0.00 ^a^	0.21 ± 0.06 ^a^	0.19 ± 0.02 ^a^	0.20 ± 0.01 ^a^	0.20 ± 0.06 ^a^	0.20 ± 0.01 ^a^	0.21 ± 0.02 ^a^	0.21 ± 0.03 ^a^
EA-2/(mg/g)	0.07 ± 0.02 ^a^	0.07 ± 0.02 ^a^	0.07 ± 0.01 ^a^	0.07 ± 0.00 ^a^	0.07 ± 0.02 ^a^	0.09 ± 0.00 ^a^	0.06 ± 0.00 ^a^	0.08 ± 0.00 ^a^
PA/(mg/g)	0.69 ± 0.04 ^a^	0.65 ± 0.18 ^a^	0.63 ± 0.06 ^a^	0.61 ± 0.03 ^a^	0.65 ± 0.02 ^a^	0.62 ± 0.02 ^a^	0.69 ± 0.04 ^a^	0.67 ± 0.01 ^a^
DPPH (mg trolox/g extract)	30.80 ± 2.59 ^bc^	31.64 ± 5.06 ^b^	33.06 ± 1.52 ^a^	29.33 ± 1.84 ^c^	29.12 ± 2.25 ^c^	29.23 ± 2.36 ^c^	28.73 ± 4.32 ^cd^	26.88 ± 5.21 ^d^
ABTS^+^ (mg trolox/g extract)	32.31 ± 2.03 ^a^	32.66 ± 2.01 ^ab^	32.72 ± 1.01 ^a^	32.31 ± 1.01 ^a^	32.09 ± 3.01 ^bc^	31.68 ± 2.02 ^cd^	32.41 ± 2.04 ^ab^	31.46 ± 1.01 ^d^

Note: Dehydrotumulosic acid (DTA-1); 3-Epidehydropachymic acid (EA-1); Polyporenic acid C (PAC); 3-Epidehydrotumulosic acid (DTA-2); Dehydrotrametenolic acid (EA-2); Poric acid (PA); Duncan’s labeling method was used between the treatment groups. Peer data with the same letter (a, b, c, d) shows no significant difference or negligible difference across treatment groups (*p* > 0.05), while excluding the same letter indicates a significant difference (*p* < 0.05); HD means hot-air drying, SD means integrated shade drying, and ID means conventional infrared drying.

## Data Availability

The data that support the findings of this study are available from the corresponding author on request.

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
