# Peer review of "Changes in the Volatile Flavor Substances, the Non-Volatile Components, and the Antioxidant Activity of *Poria cocos* during Different Drying Processes"

_molecules, 2024, doi:10.3390/molecules29194777_

Round 1
Reviewer 1 Report
Comments and Suggestions for Authors
Thank you for this nice paper, very interesting and inspiring. I will attach a pdf version in order to help you to address the suggested modifications and also hire you have additional comments.
Major questions addressed:
The authors have developed a study on the modifications of compositions
measured after usual drying processes of Poria cocos. Different
procedures (eight) have been performed and compared in order to suggest
the best one (variation of temperatures, air drying and infrared
drying). They have analysed proportions of volatile derivatives, but
also amino-acids, high-weight compounds like proteins and
polysaccharides, with triterpenoids ingredients and alcohol-soluble
fractions. Related to these new compositions, they have studied the
antioxidant properties of the resulting mixtures. The reader understand
that they have limited the biological properties to the anti-oxidant one
(this test is the easiest one), and hope that others will be addressed
in the future. In other terms, it will be useful to suggest « one
preferred process » for « one biological property ».
Originality / Particular gap
The major originality of this work relies on the discovery of the fact
that the process will strongly modify the composition and subsequently
the aroma and activity properties, suggesting to the people interested
by Poria cocos to carefully select the best process. Regarding the
various anti-inflammatory, anti-oxidant, anti-tumoral and hypoglycemic
effects, it appears now clear that for a particular targeted property,
the best process should be selected, depending on the expected
compositions. The most interesting part of this article concerns the
correlations that the authors have established between the different
clusters, arranging clusters to help the understanding of compositions.
Comparison to the literature
The state of the art appears solid and clearly explained, especially on
the extracting and drying methods other authors have used. This work
extends the approaches and compares the different methods, correlating
them with the various obtained compositions. According to the authors
statement, no other study exists on the changes of compositions related
to the drying process of Poria cocos.
Suggested modifications / Methodology, controls
The identification of 47 different components is clearly detailed and
convince the reader that this study is very serious, even if there is
undoubtedly more substances present in these natural mixtures.
GC-MS seems to be a perfect analytical approach, for volatile
derivatives. It will be interesting in future studies to complete the
collection and identification of compounds (maybe with robust apparatus
or with more comparisons of minor components). Concerning the
amino-acids (16 followed very precisely) and triterpenoids composition,
the choice of automated analyzer and HPLC are the best one, especially
because it is possible to compare with authentic samples. For the
proteins (Kjeldahl) and polysaccharides, only « crude methods » could be
performed. We can think that it would have been useful to characterize
them more precisely, but complexity of these mixtures is a clear limit
of this method, and to the best of our knowledge, it is really difficult
to have a better result for identifications.
Conclusions in relation with work / disclosed arguments
The conclusion is short, but summarizes efficiently the highest points
of this rich paper.
References and bibliography
The state of the art seems complete and references cited in the text are
correctly indicated.
Comments on tables and figures
The radar chart described in figure 1 is very useful to observe the
different compositions. The values in tables 1 and 2 are very precise
(for amino-acids and proteins / and terpenoids) and the correlations
between volatile and non-volatiles species, in figures 2 and 4, with
organised clusters, are very interesting to see the evolutions and
relationships.
Figure 5 is too small, but we suggest that it will possible to expand it
in supplementary material.
Best regards

Author Response
Thank you very much for taking the time to review this manuscript. You can find detailed modification details and corresponding highlights in the resubmitted file.
Comments 1: Figure 5 is too small, but we suggest that it will possible to expand it in supplementary material.
Response 1: We have put the clear Figure 5 in the supplementary material.
Comments 2: I will attach a pdf version in order to help you to address the suggested modifications and also hire you have additional comments.
Response 2: Thanks for your careful checks. We are sorry for our carelessness. In the peer-reviewed pdf file uploaded by you, we have modified it according to the modification opinions and marked it in red in the revision file uploaded by us. At the same time, other changes are also marked.
We have a question that needs to be specifically addressed : In line 220 of “total triterpenoids, triterpenoids”, we want to express the meaning of the total triterpenoids and triterpenoids (similar to the relationship between whole and part); we are not clear, and very sorry may have caused you trouble. (In this part, we marked it with green.)
Reviewer 2 Report
Comments and Suggestions for Authors
This interesting study shows that the aroma, active components, and activity of Poria cocos can be affected by drying methods. The authors conclude that the HD70 samples (hot air drying at 70℃) had the highest content of polysaccharides, triterpenoid ingredients, alcohol-soluble extracts, and antioxidant activity.
- Please explain in the text what the shen-nong-ben-cao-jin is.
-The introduction and/or Methods section should include information regarding drying methods used in the study and the differences between them.
- Please arrange the reference format. References must be numbered in order of appearance in the text. In the text, reference numbers should be placed in square brackets.
- Table and Figure legends (where applicable). The authors stated, “Data points in the same row with different letters are significantly different (p < 0.05)”. It is unclear what comparisons (between what samples?) have been made. Please rewrite and explain since it is crucial to understand the results obtained adequately.
Comments on the Quality of English Language- The manuscript needs minor revisions in language and grammar.
Author Response
Thank you very much for taking the time to review this manuscript. You can find detailed modification details and corresponding highlights in the resubmitted file.
Comments 1: Please explain in the text what the shen-nong-ben-cao-jin is.
Response 1: We think this is an excellent suggestion. We have explained shen-nong-ben-cao-jin in the article, page 1, lines 27-29.
Comments 2: The introduction and/or Methods section should include information regarding drying methods used in the study and the differences between them.
Response 2: Thank you for your comment. In the introduction, we added information about the drying methods in the study and the differences between them, pages 1-2, lines 40-46.
Comments 3: Please arrange the reference format. References must be numbered in order of appearance in the text. In the text, reference numbers should be placed in square brackets.
Response 3: Thank you very much for noticing the format of our references. We have made corrections and the serial number of the references in the article has also been updated.
Comments 4: Table and Figure legends (where applicable). The authors stated, “Data points in the same row with different letters are significantly different (p < 0.05)”. It is unclear what comparisons (between what samples?) have been made. Please rewrite and explain since it is crucial to understand the results obtained adequately.
Response 4: Thank you for the suggestion. We' re sorry we didn't write it clearly. We've added the complete information below the chart, pages 6 and 9, lines 176-177 and 211-213.
Comments 5: The manuscript needs minor revisions in language and grammar.
Response 5: We sincerely appreciate your suggestion. We will pay close attention to details such as spelling, punctuation, and sentence structure to ensure that the revised manuscript is of higher quality in terms of language expression.
Reviewer 3 Report
Comments and Suggestions for Authors
Authors investigate the effect of different drying methods on the composition of Poria cocos bioactive compounds, I think the publication must be improved,
Mayor points: the introduction is short, authors mention that polysaccharides, triterpenoinds and amino acids are responsible of the bioactive properties, this information is very general, they could describe which type of polysacharide, triterpenoinds and amino acids are found in P.cocos and what is the specific biological activity associated.
In results and discussion, as expected, there is degradation and changes in volatile compounds because of the drying method (temperature), however authors may discuss or predict how the composition changes could affect the biological properties of P. cocos, according to the information provided in introduction (type of molecule/specific biological property).
Minor points: the specie name Poria cocos is not written correctly, starting with the title, in must be in italics, I think after one time is written Poria cocos in italics, then it must be abbreviated as P. cocos.
Some mistakes in last names (lines 43 and 138) and references, they are in capital letters
Author Response
Thank you very much for taking the time to review this manuscript. You can find detailed modification details and corresponding highlights in the resubmitted file.
Comments 1: Mayor points: the introduction is short, authors mention that polysaccharides, triterpenoinds and amino acids are responsible of the bioactive properties, this information is very general, they could describe which type of polysacharide, triterpenoinds and amino acids are found in P.cocos and what is the specific biological activity associated.
Response 1: Thank you for your comment. We also think this is necessary, so we added this section to the introduction, which specifically describes the biological activities of alkali-soluble polysaccharides, triterpenoinds and amino acids, page 2, lines 53-56.
Comments 2: In results and discussion, as expected, there is degradation and changes in volatile compounds because of the drying method (temperature), however authors may discuss or predict how the composition changes could affect the biological properties of P. cocos, according to the information provided in introduction (type of molecule/specific biological property).
Response 2: Thank you for the suggestion. In the results and discussions, we added the changes of D-limonene and 2-undecanone content with the change of drying temperature and the relationship with Poria cocos, hoping that readers can more clearly understand how the changes of volatile components affect the biological characteristics of P. cocos, page 2, lines 86-90.
Comments 3: Minor points: the specie name Poria cocos is not written correctly, starting with the title, in must be in italics, I think after one time is written Poria cocos in italics, then it must be abbreviated as P. cocos.
Response 3: Thank you for the suggestion. The “Poria cocos” in the title has been changed to italics. Poria cocos in the text has all been replaced by P. cocos.
Comments 4: Some mistakes in last names (lines 43 and 138) and references, they are in capital letters.
Response 4: Thanks for the suggestion. The format of the references in the text has been updated and numbered in the order in which they appear in the text. In the text, the reference numbers are placed in square brackets.
Round 2
Reviewer 3 Report
Comments and Suggestions for Authors
Authors improve the manuscript